# Antibacterial Optimization of Highly Deformed Titanium Alloys for Spinal Implants

**DOI:** 10.3390/molecules26113145

**Published:** 2021-05-24

**Authors:** Katarzyna Kasperkiewicz, Roman Major, Anna Sypien, Marcin Kot, Marcin Dyner, Łukasz Major, Adam Byrski, Magdalena Kopernik, Juergen M. Lackner

**Affiliations:** 1Institute of Biology, Biotechnology and Environmental Protection, Faculty of Natural Sciences, University of Silesia in Katowice, 40-032 Katowice, Poland; 2Institute of Metallurgy and Materials Science, Polish Academy of Sciences, 30-059 Cracow, Poland; a.sypien@imim.pl (A.S.); l.major@imim.pl (Ł.M.); a.byrski@imim.pl (A.B.); 3Faculty of Mechanical Engineering and Robotics, AGH University of Science and Technology, 30-059 Cracow, Poland; kotmarc@agh.edu.pl; 4Faculty of Science and Technology, Jan Dlugosz University in Czestochowa, 42-200 Czestochowa, Poland; m.dyner@ajd.czest.pl; 5Faculty of Metals Engineering and Industrial Computer Science, AGH University of Science and Technology, 30-059 Cracow, Poland; kopernik@agh.edu.pl; 6Institute of Surface Technologies and Photonics, Functional Surfaces, Joanneum Research Forschungsges.m.b.H., 8712 Niklasdorf, Austria; juergen.lackner@joanneum.at

**Keywords:** multilayer coatings, antimicrobial materials, biomaterials, Ag nanoparticles, titanium, spinal implants, biofilm, SAOS2, cytotoxicity

## Abstract

The goal of the work was to develop materials dedicated to spine surgery that minimized the potential for infection originating from the transfer of bacteria during long surgeries. The bacteria form biofilms, causing implant loosening, pain and finally, a risk of paralysis for patients. Our strategy focused both on improvement of antibacterial properties against bacteria adhesion and on wear and corrosion resistance of tools for spine surgery. Further, a ~35% decrease in implant and tool dimensions was expected by introducing ultrahigh-strength titanium alloys for less-invasive surgeries. The tested materials, in the form of thin, multi-layered coatings, showed nanocrystalline microstructures. Performed direct-cytotoxicity studies (including lactate dehydrogenase activity measurement) showed that there was a low probability of adverse effects on surrounding SAOS-2 (*Homo sapiens* bone osteosarcoma) cells. The microbiological studies (e.g., ISO 22196 contact tests) showed that implanting Ag nanoparticles into Ti/Ti_x_N coatings inhibited the growth of *E. coli* and *S. aureus* cells and reduced their adhesion to the material surface. These findings suggest that Ag-nanoparticles present in implant coatings may potentially minimize infection risk and lower inherent stress.

## 1. Introduction

Orthopedic spine surgeries are highly challenging interventions with a high risk of neurological damage. In pediatrics (in 3% of adolescent), the focus is on the orthopedic stabilization of kyphosis and scoliosis, while fusions for osteoporotic vertebrae and spinal tumor resections dominate in the case of elderly patients. The main problems in all orthopedic spine surgeries with screw–rod–plate systems are bacterial surgical site infections (SSI), with rates up to 41.7% of surgeries in pediatrics (especially in myelodysplasia), resulting in patient morbidity, multiple operations and risks of paralysis. Especially low-vascularized sites around spinal implant surfaces are prone to SSI by *Staphylococcus aureus*, *Staphylococcus epidermidis*, *Pseudomonas aeruginosa* and *Propionibacterium acnes*. Immediate infection risks have been minimized by surgical techniques, but a reduction in delayed SSIs after a subclinical quiescent period is still a pressing need. They are generally diagnosed in late, highly intense stages. A statistical peak for such delayed SSIs occurs 7–8 weeks after surgery. Due to the formation of surface protein layers under physiological conditions, the titanium is highly biocompatible both with cells and bacteria [1,2,3]. Bacterial adhesion and colonization on titanium implants form biofilms, in which adherent bacteria are protected from host defense systems and bactericidal agents [3,4,5]. Additionally, local body defense is severely disturbed by surgical trauma in the early phases after implantation as well as compromised on account of a small number of blood vessels in these areas after healing. Consequently, antimicrobial strategies must precede bacterial colonization and be introduced to the most endangered areas at high, but non-cytotoxic, doses.

In the past decades, extensive research and development has been performed on various strategies for antibacterial bone-contacting implant surfaces, including antibiotic-loaded coatings with Gentamicin (an aminoglycoside) and Vancomycin in porous calcium phosphate, hydroxyapatite, sol-gel and polymer matrices (see overview in [6]). Ceramics (bulk and coatings) are often loaded with drugs through physical absorption obtained by dipping or immersion, which results in burst release in less than one hour [7,8]. Covalent bonded antibiotics decrease the drug release rate or provide antibacterial protection in direct contact with bacteria [9,10]. Further, sol-gel and bio-resorbable polymer coatings (Polylactic acid—PLA, Polyglycolic acid—PGA, Polyethylene glycol—PEG) are useful for achieving slower release rates [11]. Nanopore and nanotube surfaces enable antibiotic drug loading (e.g., by infiltration or lyophilization) and merge it with high osteoconductivity for improved bone integration, but they all struggle with too-high drug release rates [12,13]. In conclusion, long-term antibiotic delivery at effective concentrations (above the minimum for preventing formation of bacterial resistance and below the cytotoxicity threshold) is a challenging problem, requiring a novel approach like the hybrid materials concept.

Alternatives with a lower risk of drug resistance to antibiotics, but weighted with an eventual danger of in vivo cell damage [3,14], are coatings containing non-antibiotic organic agents (e.g., chlorhexidine/chloroxylenol adsorbed in titania [15,16]). The most-studied surfaces are, by far, those based on cation-functionalized polymers such as quaternary polyethylenimines (QPEIs) or polymers with cationic quaternary ammonium salts (QAS) [17,18]. Quorum-sensing (QS) inhibitors and peptides are also intensively studied, but of interest are only those with low antimicrobial activity against a broad spectrum of biological species and the tendency to develop drug resistance [19,20]. In contrast, non-antibiotic, inorganic Ag has a broad antibacterial spectrum at very low ppb (parts per billion) concentrations in tissue [21,22,23,24,25]. It inhibits bacterial adhesion [26], as well as having both a long-lasting effect and low cytotoxicity at necessary doses [27], backed by very low risk of resistance development to bacteria. Beyond others, Ag, Ti-Ag [24], TiN-Ag [28,29], TiN-Ag [30,31], Ag-DLC [32,33], and Ti-Mg-Ag(Pt) [34] coatings are presently in development, or, i.e., under clinical study. Numerous techniques of incorporating Ag nanoparticles into coatings are being examined: e.g., radio-frequency magnetron sputter-deposition [35]. The drawback of Ag is its release kinetics to surrounding tissue, which is electrochemically dependent on the pH value and may be too low to counteract bacteria colonization in the injured surrounding tissue.

Resistance to bacterial adhesion can be found also in other coatings; i.e., TiO_2_ in combination with UV light may photocatalyze hydrocarbons and bacteria [36], or TiNO_x,_ with especially ~104 µΩ cm electrical resistance, minimizes bacterial adhesion, too [37]. The same effect could be achieved with protein-resistant poly(methacrylic acid) and poly(ethylene glycol) [23,38], which both significantly reduce bacterial adhesion (e.g., brush-like 2D structure pattern of PEG), but hinder osteoblast adhesion. In contrast, bio-active chitosan and hyaluronic acid may stimulate differentiation of osteoprogenitor cells [39,40]. Finally, nitric oxide (NO) release from sol-gel coatings acts anti-microbial in vitro [41]. Nevertheless, all these films have low mechanical stability and adhesion due to grafting to Ti by linking molecules [40,42] or embedding the bioactive molecules in layer-by-layer self-assembled coatings.

Finally, bacterial adhesion is dependent on the protein layer on the biomaterial surface on which fibrinogen is counted as one of the major plasma proteins. As well as fibronectin, mainly involved in blood clotting due to the adhesion of *S. aureus* and *E. coli* [43,44,45], while adsorbed albumin on Ti implants drastically reduces *S. aureus* and *P. aeruginosa* adhesion [46]. In contrast, *S. epidermidis* adheres and colonizes surfaces using a self-generated, viscous biofilm composed of polysaccharides [47], demanding anti-adhesive, hydrophobic surface properties of biomaterials.

The use of silver in coatings may seems to be a standard approach in the case of materials presenting antibacterial properties. However, one of the main problems associated with biocompatibility and biotolerance is the toxicity to human tissues of the materials concerned and the reaction of the immune system. In some cases, the problem is so specific that it manifests itself as a strong allergic reaction. Silver nanoparticles are generally considered to be useful antibacterial agents. even as their use must be always carefully monitored. Therefore, in the present work, it has been decided to use them as a coating enhancer, only.

An ultrafine (<1 μm) or even nanocrystalline (<100 nm) material prepared by a severe plastic deformation (SPD) process is characterized by very high strength, which allows for lightening the implants. SPD is a metal forming process in which a metal is subjected to an ultrahigh plastic strain in order to refine its grain structure increasing both its strength and ductility. The dominating SPD technique is equal channel angular pressing (ECAP), which enables up-scaling the SPD to an industrial-applicable size. During ECAP processing, the raw material is pressed through a die, with a channel bent at a predetermined angle (usually 90° or 120°). ECAP process enables introducing a large amount of simple shear deformation to enhance hydrostatic pressure in single or multiple processing steps without changing the cross-section of the work piece. Due to this large plastic deformation, ECAP processing of Ti and Ti alloys has resulted in an UFG (ultrafine-grained) structure with a high defect density (dislocations, grain boundaries) and correspondingly increased strength. That is why the titanium alloys which undergo SPD should be a material of choice for products as demanding as spine implants.

The goal of the work was to elaborate multilayer coatings dedicated to spine implants with microstructures ensuring both low cytotoxicity and microbiological impact.

## 2. Results

### 2.1. Fatigue Properties of the Strengthened Substrate

Fatigue properties of CP-Ti (commercially pure titanium) and Ti64, under conventional and ECAP heat treatment, were investigated. Plastic-strain-controlled low-cycle fatigue (LCF) tests—according to ASTM E606—in the range of 10^3^ to 10^5^ (1000–100,000) cycles at a constant plastic strain rate of 1 × 10^−^^3^ s^−1^ and at variable frequencies (f = 0.05–0.906 Hz) were performed. Additionally, stress-controlled high-cycle fatigue (HCF) testing (according to ASTM E466) in the range of 10^5^–10^7^ (100,000–10,000,000) cycles at a constant frequency (f = 10 Hz) was carried out. Manufacturing of special fatigue samples was ordered at Westmoreland Mechanical Testing & Research, Ltd.

### 2.2. Microstructure Analysis of the Smart Bioactive Coating

The coatings were applied on both the CP-Ti and austenitic steel substrates. They differed in their content of Ag nanoparticles and thickness. The detailed microstructure characteristics were carried out on the thickest coating (~3.80 μm) containing the largest amount of Ag nanoparticles, which allowed the best description of this type of structure (Figure 1A).

The coating was composed of two parts. The inner part (the first from the substrate) was the proper Ti/Ti_2_N multilayer structure implanted with nanoparticles of Ag (Figure 1). The presence of the latter was confirmed by phase analysis carried out by means of electron diffraction (SAEDP). Both the metallic Ti (thinner) and ceramic Ti_2_N (thicker) layers were built of columnar crystallites. Retaining of the same diffraction contrast across multilayer boundaries indicated establishing a particular crystallographic orientation between Ti and Ti_2_N phases. In order to better illustrate the distribution of Ag nanoparticles in this part of the coating, both in the metallic and ceramic layers, observations were made using the so-called dark field (TEM DF) technique (Figure 1B). It allowed to differentiate nanoparticles as white and black dots against the columnar crystallites (see: black arrows).

The presence of nanoparticles in the Ti/Ti_2_N structure was independently confirmed by high resolution TEM (HRTEM: high-resolution transmission electron microscopy) observations (Figure 2).

Qualitative analysis of the chemical composition was carried out by means of X-ray energy dispersive spectroscopy (EDS) from the area marked with red square (Figure 2A) and along the line, perpendicular to the coating/substrate boundary (Figure 2B). The performed measurements indicated that the presence of lighter mass–thickness contrast at the near-surface area was associated with a much lower content of both titanium and silver.

### 2.3. Mechanical Properties

The tests were carried out for up to 2000 cycles. The wear rate was calculated according to the recommendations of ISO 20,808 from dependencies:(1)$$W=\frac{V}{F_{n}\cdot S}$$

In the formula, *W* is the wear indicator in mm^3^·N^−^^1^ m^−^^1^*, V* is the volume of the material removed, *F_n_* is the normal load and *S* is the overcome friction path.

The coefficients of the friction and wear indicators were determined according to ISO standard 20808:2016. Tests of the friction coefficients and wear indicators were performed based on ISO 20808:2016 (Table 1).

### 2.4. Direct Cytotoxicity Analysis

Direct cytotoxicity tests were conducted for Ti/Ti_x_N materials implanted with Ag nanoparticles. The results of the tested materials are shown in Figure 3. As a reference, a glass plate was taken.

The obtained results (Figure 3A–E) document a relatively good cell survival rate on the surface. Compared to the reference material (Figure 3A), the most necrotic cells were found for Ti/TiN + 10% Ag (Figure 3D). This is shown by the increased number of red spots on the image, which correspond to the excitation of propidium iodide (PI), which marks necrotic cells. High concentrations of Ag nanoparticles lead to the formation of Ag aggregates, resulting in lowering the concentration of biologically active Ag nanoparticles. It indicates that a higher concentration of nanoparticles may show lower cytotoxicity (Figure 3E) [48]. No increased number of necrotic cells was found for the Ti/TiN + 5% Ag coating (Figure 3B). However, this material was not conducive for their growth. For Ti/TiN + 7.5% Ag (Figure 3C) and Ti/TiN + 15% Ag (Figure 3E), a large number of cells with active mitochondria and low mortality of cells were found. This proves good biological properties of the tested materials.

### 2.5. Lactate Dehydrogenase

In order to evaluate the cytotoxicity of examined coatings, the extracellular lactate dehydrogenase activity of the SAOS-2 cell line was measured. Results are shown in Figure 4.

The results (presented as the average of five independent iterations) of cytotoxicity tests using the lactate dehydrogenase method showed the worst properties for Ti/TiN + 5% Ag coatings. As the silver content in the coating structure increased, improved coating properties were observed. This was manifested by a reduced risk of destructive effects of the materials on the surrounding cells.

### 2.6. Surface Interaction of Coatings with Selected Strains of Bacteria and Fungi

#### 2.6.1. Growth, Adhesion and Bacterial Biofilm Formation on the Surface of Biomaterials

The biomaterials under examination were incubated with cultivation of *Pseudomonas aeruginosa*, *Streptococcus pyogenes* and *Candida albicans*, and an increase in OD600 (optical density) levels was observed for *Streptococcus pyogenes* and *Candida albicans* in cultivation with the tested biomaterials. This may have indirectly indicated the growth of microorganisms during the incubation as well as implied that the biomaterials had neither bactericidal nor bacteriostatic properties. Bacterial adhesion on the tested surface was observed using the SEM (scanning electron microscopy) technique. The results are presented in Figure 5.

Adhesion of single bacterial cells in the shape of small, short rods was observed on the surface of tested materials. Cells of *Pseudomonas aeruginosa* formed loose clusters (Figure 5A). No formation of biofilm structure by this microorganism was observed. On the surface of the tested material (Figure 5B), few spherical cells forming characteristic chains were visible. Bacterial adhesion on this material was very low. The adhesion of numerous *Candida albicans* cell was observed on the surface on the tested material (Figure 5C). Spherical cells (typical for these microorganisms) as well as pili forms (elongated structures of varying length and branched structures composed of pudding cells) were visible. The filamentous cells presented the structure of pseudomycellium (Figure 5C).

#### 2.6.2. Tests of Viability of Microorganisms on the Tested Biomaterials

The study aimed to determine the cytotoxic activity of biomaterials based on cell membrane integrity analysis because all permanent damage to the cell membrane leads to bacterial cell death [49]. In the test used to determine changes in cell integrity, the cells of the *Pseudomonas aeruginosa* bacterium were incubated with a positive charge dye solution (fluorescent propidium iodide). The results are presented in Figure 6.

For the bacterium *Pseudomonas aeruginosa*, a large number of dead bacterial cells for the Ti/TiN + 15% Ag coating were found in the tested cultures with the biomaterials, which confirmed its bactericidal properties (Figure 6E). In comparison with the results of direct cytotoxicity, the presently obtained results are proof of the very good antibacterial properties of this material.

#### 2.6.3. Test of Antimicrobial Activity of Biomaterials According to ISO 22196

In order to determine the antimicrobial activity of a material, the difference in microorganism growth on treated and untreated surfaces was determined. The numbers of recovered bacteria cells (*S. aureus*) from each specimen are shown in Figure 7.

Diversity in the number of recovered *Staphylococcus aureus* cells after 24 h of incubation was observed. The highest number (10^6^ CFU (Colony Forming Units)) of viable cells were obtained from untreated samples. Both Ti/TiN + 7.5% Ag and Ti/TiN + 10% Ag showed moderate bacteriostatic activity, whereas Ti/TiN + 15% Ag and Ti/TiN + 5% Ag showed bactericidal activity—no viable *S. aureus* cells were recovered from specimens. Due to the Log scale, the standard deviation for Ti/TiN + 10% was quite small.

Material yields antibacterial properties if the obtained R value is greater than 2, as stated in ISO 22196. Every tested biomaterial beyond Ti/TiN + 7.5% Ag showed antibacterial activity against Gram-positive *Staphylococcus aureus*. Furthermore, the greatest Log reduction (6 orders of magnitude) for Ti/TiN + 15% Ag and Ti/TiN + 5% Ag compared to untreated specimens was observed.

## 3. Discussion

The present experiment covered deposition of multi-layer Ti/Ti_x_N coatings implanted with Ag nanoparticles and their detailed microstructural characteristics. This multilayer coating was elaborated with the aim of increasing not only the mechanical properties of the surface, but also its antibacterial properties. The performed investigations using the TEM method showed an even distribution of nanoparticles in the multilayer structure. Various mechanical tests allowed for selecting the coating presenting the highest wear resistance. On the basis of the results obtained from the experiments with the first group of coatings produced on spinal implants, an improved series of coatings on spinal implants was produced. They were built from two parts; i.e., the first one from the substrate was made of a Ti/Ti_2_N multi-layer implanted with 7.5% at. Ag, while the second was based on a-C:H implanted with 3 and 7% at. Ag and 5% at. Si. The presence of nanoparticles in the a-C:H part was associated not only with increasing the antimicrobial properties of coatings, but also with lowering the inherent stress.

The improvement in mechanical properties resulting in improved microbiological properties most likely resulted from the elimination of structural defects by anchoring silver nanoparticles. The magnetron method is an application technique based on the physical impingement of disc particles in a protective atmosphere. Diffusion does not occur in this technique; it is only considerable as a pseudo-diffusion phenomenon.

The antimicrobial properties of Ti/Ti_x_N coatings implanted with Ag nanoparticles were tested according to the ISO 22,196 norm. This norm is widely utilized in biomaterial engineering as a simple and reliable means of verifying anti-pathogenic properties of surfaces.

*Staphylococcus aureus* and *Escherichia coli* are responsible for most implant-related infections; therefore, usage of these bacteria in experiments is justified, similar to results presented elsewhere [50]. Bacterial infection is always established with adhesion of the cell to the surface of tissue or biomaterial, followed by bacterial growth and biofilm formation. Inhibiting infection during one of those steps prevents spreading of the SSI and decreases coating bio-corrosion. Ag nanoparticles are known for their antibacterial properties, and several mechanisms have been raised. In particular, they generate reactive oxygen species that may disrupt cell wall synthesis, inhibit enzymes and corrupt DNA replication. Thereby, biomaterials which release Ag nanoparticles may significantly reduce the number of bacteria adhering to a surface and prevent them from growing and spreading further [51]. Carried-out tests indicate that specimens Ti/TiN + 15% Ag and Ti/TiN + 5% Ag stand out in their bacteria contact-killing properties (*R* index = 6.2); however, the Ti/TiN + 10% Ag specimen showed significantly lower antimicrobial activity (*R* index = 2.2), although it was still higher than 2.0 and likewise may have been effective. The Ti/TiN + 7.5% Ag material showed only limited antibacterial properties (*R* index = 1.3). Results were partially unexpected because a higher concentration of Ag nanoparticles should result in greater *S. aureus* growth inhibition, and thus a higher *R* value [52]. As mentioned above, it could be caused by aggregation of Ag nanoparticles [48] or development of nanosilver resistance mechanisms in bacteria [53].

## 4. Materials and Methods

### 4.1. Substrate Preparation

In the work for the substrate, CP-Ti grade 4 was used. In order to fragment the structure, an extrusion in the angle channel was used, or so-called equal channel angular pressing. The process parameters of ECAP were optimized in order to reduce the number of passes needed to achieve proper mechanical properties and cost effectiveness. Up to 24 ECAP passes were carried out, and the mechanical properties were estimated. As part of post-processing, thermal treatments were performed. To further improve the mechanical properties of ECAP processing, thermal treatments were carried out. Regimes for thermal treatments and dwell times were as follows: 150–350 °C and 0.5–20 h, respectively. The plastic working and postprocessing were intended to homogenize the structures and properties. As the final substrate material, a fine-grained anisotropic material was used.

### 4.2. Smart Bioactive Coatings to Control Biological Interaction

The deposition of the coatings was performed by physical and plasma-enhanced chemical vapor deposition (PVD and PECVD, respectively) on the SPD-strengthened Ti alloys. Prior to the deposition, all substrates were polished, cleaned ultrasonically in alcohol and dried in a vacuum. The manipulation of the substrates occurred under clean room conditions to prevent any dust deposits on the surfaces. After mounting the substrates in the vacuum deposition chamber (Leybold Oerlikon, Cologne, Germany) (horizontally positioned at ~100 mm distance to the coating sources), the chamber was evacuated to a 2 × 10^−^^5^ mbar starting pressure. Prior to the deposition, a final cleaning and activation of the polymer surfaces occurred by applying the Ar-O_2_ plasma of a linear anode layer ion source. All tested materials are listed in Table 1. Magnetron sputtering was applied right after ion source-based plasma cleaning and activation. Sputtering from pure pyrolytic carbon target materials (Schunk, Vienna, Austria) occurred in an N_2_ + Ar atmosphere to obtain titanium nitride coatings. All used gases were of 5N purity. Various deposition times helped to obtain different thicknesses. Three types of coatings differing in the nanoparticle contents of the outer a-C:H part of the coating were produced (Table 2).

#### 4.2.1. Microstructure

Microstructure characterization was carried out using scanning and transmission electron microscopy (SEM and TEM, respectively) as described in detail elsewhere [54]. It was performed with the help of a FEI Quanta 3D FEG Scanning Electron Microscope and a Tecnai G2 F20 (200 kV) FEG, located in the Accredited Testing Laboratories at the IMMS, PAS.

#### 4.2.2. Mechanical Properties of the Thin Films

The mechanical properties were estimated by measuring the micro-hardness by applying the scratch test and the wear test at the contact point of the shield. The method was similar to one described elsewhere [55,56]. Indentation tests were carried out at loads of 2 mN and 5 mN and accompanying growth and deceleration speeds of 4 mN/min and 10 mN/min. A diamond indenter with Berkovich geometry was used for testing. Scratch tests were performed using a Rockwell C indenter with a radius of 200 μm. The length of the scratch test was 5 mm, and the indentation rate was 5 mN/min. The load range for all coatings was 0–30 N. Tribological tests (for wear) were made at the sphere–shield interface. The ball was made of Al_2_O_3_ with diameter *d* = 6 mm. Tests were made at 1 N load. The rotational speed was 120 rpm.

#### 4.2.3. Cytotoxicity

The investigations were based on direct cytotoxicity tests. The *Homo sapiens* bone osteosarcoma cell line (SAOS-2) was chosen because it could differentiate between osteocyte-like cells and was commonly used as an osteoblast model in various in vitro experiments [57]. The incubation was performed in a 24-well plate. For each well, 1.5 mL of culture medium was used. The cells were incubated for 24 h under cell culture conditions (37 °C, 90% humidity, 5% CO_2_). After 24 h, medium cells were removed and spotted with SAOS-2 lines plated (50,000 cells/well) onto a 24-well plate containing the test claim. After 48 h of cell culture in the control well, approximately 80% confluence was achieved and the experiment was terminated. Presented results were obtained during three independent experiments. After 1 h, 2 mL of culture medium were added to the culture and incubated under the same conditions for another 24 h. Counters of targeted, live and necrotic cards were evaluated with attached confocal microscopes using propidium iodide MitoTracker^®^ marker green, staining active mitochondria. This marker is located in the mitochondrial range from the width of the mitochondrial membrane. The propidium iodide test is one of the common methods of cytotoxicity testing. To determine mitochondria, cells are incubated with MitoTracker^®^ probes, which passively diffuse across the plasma membrane and accumulate in active mitochondria. After labeling their mitochondria, the cells can be treated with an aldehyde-based fixer when used, which must be fixed to allow further processing of the samples. Available MitoTracker^®^ probes are also retained after permeabilization with some detergents during the step stages. Propidium iodine penetrates inside the cell only when using cell membrane continuity. The dye does not penetrate into living cells, but in dead cells (when the membrane is permanently damaged and there is a loss of potential between the outer and inner side of the membrane), it penetrates inside, staining the cytoplasm or/and the nucleus. Propidium iodine penetrates inside the dead cells and intercalates with the nucleic acids, showing red fluorescence. The samples were stained with a solution (1 mg/mL) of propidium iodide for 5 min and then sluiced in PBS (phosphate buffer). Propidium iodine dyes damaged cells red by nucleating with nucleic acids. The more excitation of iodide, the greater the likelihood that cells located on the surface of the biomaterial were dead.

#### 4.2.4. Lactate Dehydrogenase

Lactate dehydrogenase (abbreviated as LDH) is an enzyme that catalyzes the conversion of lactate to pyruvate (in the presence of NAD^+^), and vice versa (pyruvate to lactate in the presence of NADH). It is an enzyme found in all cells and body fluids. LDH measurement was carried out in culture fluid supernatants after 48 h of SAOS-2 cell culture. High activity of LDH correlates with cell damage and is used as a marker of injury and disease [58]. Measurement was carried out using a COBAS INTEGRA 400 PLUS apparatus from ROCHE. The results obtained are given in so-called Wróblewski units [59] or international units (U).

#### 4.2.5. Analysis of the Surface Interaction of Coatings with Selected Strains of Bacteria and Fungi

The examined biomaterials were incubated respectively with “cubes” of 15 mL of a 24-h culture of *Pseudomonas aeruginosa*, *Streptococcus pyogenes*, and *Candida albicans* and “circles” of 4 mL of a 24-h culture of *Pseudomonas aeruginosa*, *Streptococcus pyogenes*, and *Candida albicans*. The strains were grown on the following media:*Pseudomonas aeruginosa* (ATCC^®^ 27,853 ™): TSA medium, cultivation at 37 °C; composition of the substrate: TSA 40 g per 1000 mL MP water; sterilization: autoclave 1 h, 121 °C.*Streptococcus pyogenes* Rosenbach (ATCC^®^ 19,615 ™): TSA + blood mutant, culture at 37 °C; composition of the substrate: TSA 40 g per 1000 mL MP water, sheep blood 50 mL per 1000 mL medium (enriched the medium and allowed hemolytic reactions). This was the composition of media dedicated to the culture of streptococci (identified as alpha- or beta-hemolytic, depending on the appearance of the colony on this medium); added after sterilization (autoclave 1 h, 121 °C).*Candida albicans*: medium, broth + sugar; composition of the medium: ordinary broth—13 g per 1000 mL of MP water; glucose—20 g per 1000 mL of medium; sterilization: autoclave 1 h, 121 °C.

The biomaterials were rinsed with a 3.5% aqueous solution of paraformaldehyde, three times. The viability of microorganisms on the tested biomaterials was tested by staining the samples with a solution (1 mg/mL) of propidium iodide for 5 min, followed by rinsing with PBS.

#### 4.2.6. Test for Antimicrobial Activity of Biomaterials According to ISO 22,196 (JIS Z 2801 Test for Antimicrobial Activity of Plastics)

To analyze the microbiological properties of the coatings according to ISO 22196, the samples were cut into 5 × 5 cm pieces and inoculated with a bacterial suspension of the units, forming a colony of approximately 2.5 × 10^5^–1.0 × 10^6^ mL^−1^. The *Staphylococcus aureus* (ATCC^®^ 6538P ™) strain was selected for the study. Samples were incubated for 24 h at 95% relative humidity and 37 °C. During incubation, the bacterial suspension was covered with a 4 × 4 cm PET (polyethylene terephthalate) film. The volume of bacterial suspension used thoroughly moistened the 4 × 4 cm film. After an incubation period, the biomaterial, together with the culture and foil, was transferred to 5 mL PBS buffer (10 mM, pH = 7) and then sonicated for 5 min to accurately recover bacterial cells from the surface. To obtain a homogeneous bacterial suspension, the sample was mixed (Vortex device). The next stage of the research was the preparation of a number of dilutions of cultures in the sterile plates—type 96 with MaxiSorp. From each dilution, 10 microliters of culture were inoculated on casein soy agar (CSA) solid base. Agar plates were incubated for 24 h at 37 °C. To determine the initial number of bacteria, the microorganisms were counted after being applied to the biomaterial and quickly washed away from its surface. An antibacterial activity index (*R*) was calculated in accordance with ISO 22,196 as follows:(2)$$R=U_{t}-A_{t}$$

In the formula, *R* is the antibacterial activity index, *U_t_* is the logarithm of the number of viable cells recovered from untreated specimens after 24 h of incubation, and *A_t_* is the logarithm of the number of viable cells recovered from treated specimens after 24 h of incubation.

## 5. Conclusions

The study indicates that Ti/Ti_x_N coatings implanted with Ag nanoparticles significantly inhibit the growth of selected microorganisms and reduce the adhesion to the material surface (as confirmed by Figure 5, Figure 6 and Figure 7). Additionally performed tests showed no significant cytotoxicity (Figure 3 and Figure 4) to the osteosarcoma (SAOS-2) cells. The results of performed experiments indicate that Ti/Ti_x_N multilayer coatings implanted with Ag nanoparticles should improve bio-compatible properties of surgical spinal implants, and in effect, reduce the number of SSIs observed after spine surgeries.

## Figures and Tables

**Figure 1 molecules-26-03145-f001:**
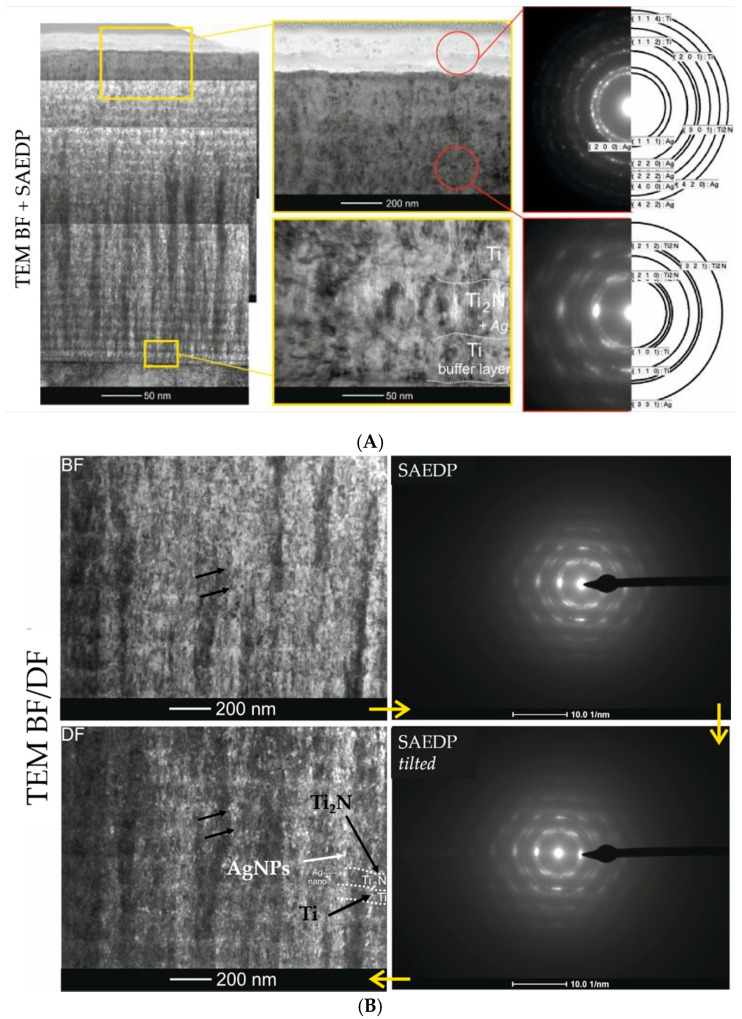
TEM/BF images and accompanying electron diffractions: (**A**) of multi-layer Ti/Ti_x_N coating with thickness (~3.80 µm) implanted with Ag nanoparticles (12.5% at.), (**B**) of Ag nanocrystallites in both Ti and Ti_2_N layers.

**Figure 2 molecules-26-03145-f002:**
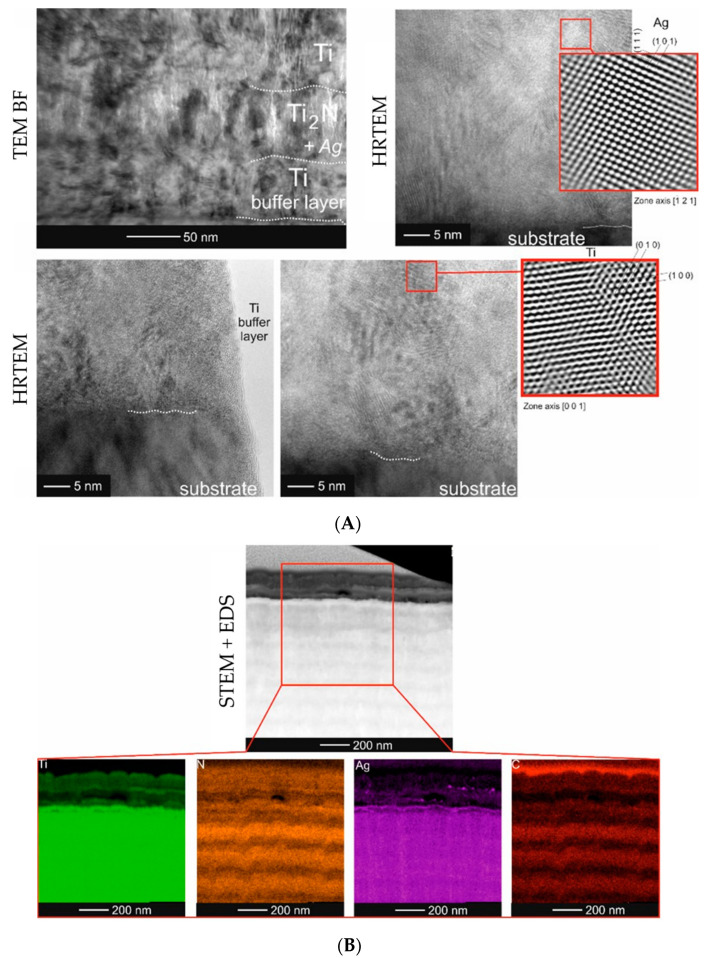
TEM/BF and HREM images of distribution of Ag nanocrystallites in Ti/Ti_2_N multilayer (**A**), STEM image and accompanying maps presenting distribution of Ti, N, Ag, C elements from area marked with square (**B**) and profiles of chemical composition across the multilayer (along the marked line) (**C**).

**Figure 3 molecules-26-03145-f003:**
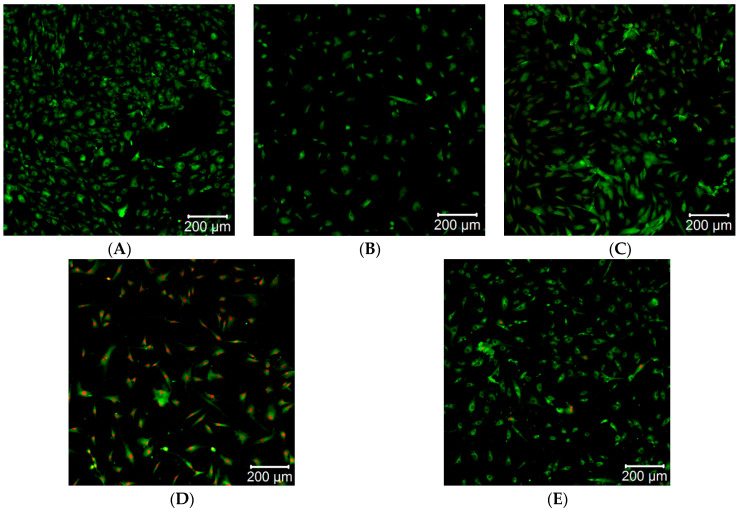
Images from confocal microscope of: (**A**) reference, (**B**) Ti/TiN + 5% Ag, (**C**) Ti/TiN + 7.5% Ag, (**D**) Ti/TiN + 10% Ag, (**E**) Ti/TiN + 15% Ag.

**Figure 4 molecules-26-03145-f004:**
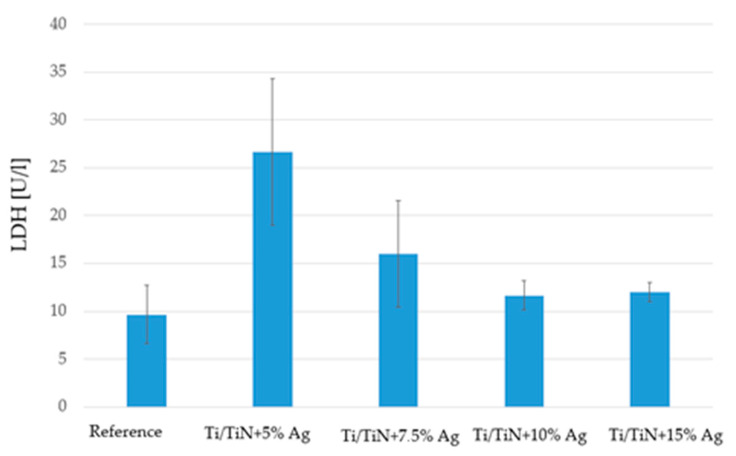
Results of cytotoxicity tests using lactate dehydrogenase (*n* = 5 ± SD).

**Figure 5 molecules-26-03145-f005:**
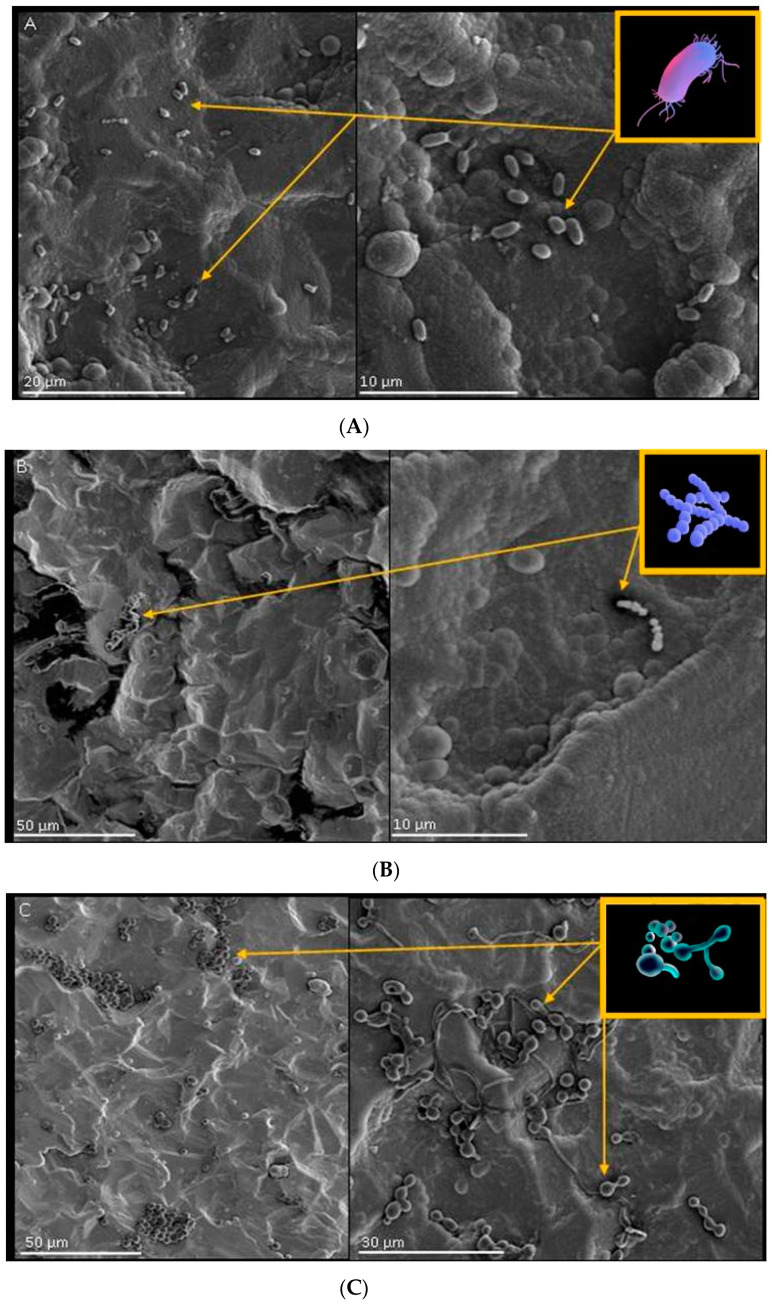
Colonization of the tested biomaterial (Ti/TiN + 7.5% Ag) by microorganisms (bacteria and fungi). Surface morphology (SEM) with the: (**A**) *Pseudomonas aeruginosa*, (**B**) *Streptococcus pyogenes*, (**C**) *Candida albicans*.

**Figure 6 molecules-26-03145-f006:**
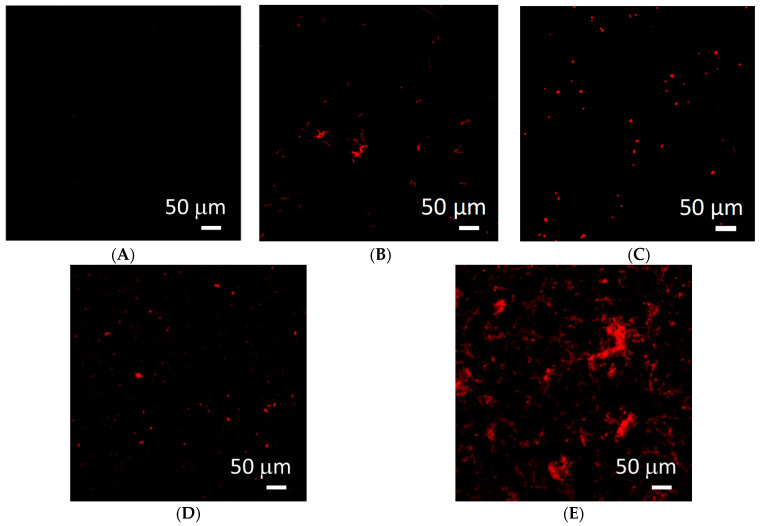
The viability of bacteria on the surface of the tested biomaterials. (**A**) Reference substrate, (**B**) Ti/TiN + 5% Ag, (**C**) Ti/TiN + 7.5% Ag, (**D**) Ti/TiN + 10% Ag, (**E**) Ti/TiN + 15% Ag.

**Figure 7 molecules-26-03145-f007:**
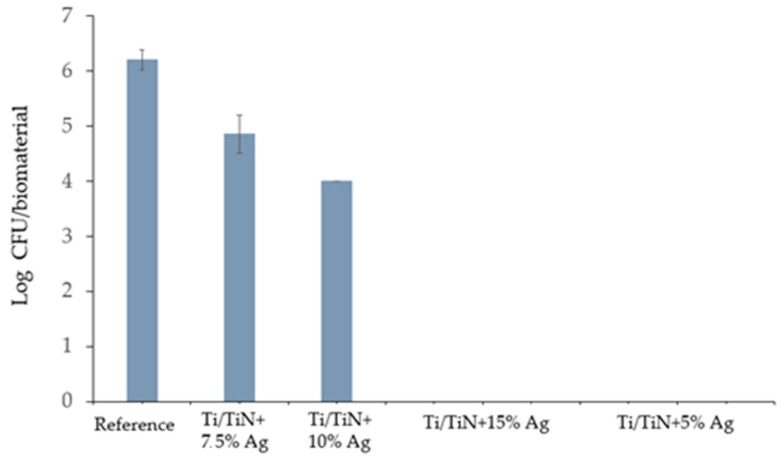
Number of recovered *S. aureus* cells from each surface after 24 h contact test (*n* = 4 ± SD).

**Table 1 molecules-26-03145-t001:** Results of friction coefficient and wear indicator tests.

Layer	Wear Rate *F* = 0.5 N *N* = 2000c × 10^−^^6^	Friction Coefficient
Ti/TiN + 5% Ag	251.7	0.428
Ti/TiN + 7.5% Ag	564.8	0.494
Ti/TiN + 10% Ag	574.1	0.478
Ti/TiN + 15% Ag	835.7	0.404

**Table 2 molecules-26-03145-t002:** The list of coatings used in experiments.

L.P.	Layer	Thickness [µm]
1	Ti/TiN 1:1 doped 5% Ag	3.40
2	Ti/TiN 1:1 doped 7.5% Ag	3.40
3	Ti/TiN 1:1 doped 10% Ag	2.70
4	Ti/TiN 1:1 doped 15% Ag	3.50

## Data Availability

The data presented in this study are available on request from the corresponding author.

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
