# Peer review of "Antibacterial Optimization of Highly Deformed Titanium Alloys for Spinal Implants"

_molecules, 2021, doi:10.3390/molecules26113145_

Round 1

Reviewer 1 Report

Comments
“Antibacterial optimization of highly deformed titanium alloys for spinal implants” by Katarzyna Kasperkiewicz et al. Interesting paper but needs to be bolstered by more in vitro data presentation. Authors should probably add more data to strengthen this work.

1)    Literature references in introduction are too old. Most of them are 10 years ago.
2)    The specific surface area of Ag plays an important role in the development of both antibacterial and biocompatibility. It is necessary to mention the particle size of Ag incorporated in each coating.
3)    In Figure 5, Why did you use Pseudomonas aeruginosa, Streptococcus pyogenes, and Candida albicans? What were biomaterials used in this test? How did you distinguish the biofilm formation and the bacterial colonization from SEM images.
4)    In Discussion, please explain more lines 276-279.
5)    In 2.4. Direct cytotoxicity analysis, why did you use the glass plate as a reference? A microscopic image of the SAOS-2 sample without Ag should be displayed. The same is true for the 2.5, 2.6.2, and 2.6.3 sections. In addition, the sample with lowest Ag content exhibited the highest cytotoxicity, this result is very interesting. Please show us the cell viability on each samples and Ag ions release behavior from those. 
6)    Is the antibacterial mechanism of this coating really contact killing? The antibacterial mechanism is usually determined by the ion elution test or the disc diffusion method. 

Author Response

All remarks given by Reviewer 1 were taken under the consideration. All parts that were inserted as new according to the reviewer's suggestion are marked in green in the manuscript. Detailed explanations are included in the attached files

Reviewer 2 Report

  1. I would suggest the authors to better describe the surface modification strategies using alternative approaches to prepare biocompatible coatings with antibacterial properties, e.g. silver nanoparticles are extensively studied by prof. Epple et al. in case of their combination with HA-based coatings, e.g. doi: 10.1016/j.apsusc.2014.12.153
  2. It is also a challenge to provide sufficient content of silver to provide both antibacterial effect and biocompatibility. How did the authors optimize the content of silver using their approach.
  3. 1 Please improve resolution and increase the fonts. It is barely seen what is depicted.
  4. I would recommend providing statistical analysis in all the results concerning biological experiments.
  5. I would suggest briefly describe if TiN coatings will not increase stress-shielding effect, since Young’s modulus is significantly higher than that of pure Ti.

Author Response

All remarks given by Reviewer 1 were taken under the consideration. All parts that were inserted as new according to the reviewer's suggestion are marked in green in the manuscript. Detailed explanations are included in the attached files.

Round 2

Reviewer 1 Report

To avoid confusion for the reader, it is recommended to remove the description of biofilm formation from 2.6. Further experiments are needed if you want to mention biofilm formation.

Author Response

Following the recommendation of Reviewer (Report Round 2), the authors removed the description of biofilm formation from 2.6 (sentence from lines 227-228). As the Reviewer aptly mentioned, the description may be confusing for the reader. The performed examination is not sufficient to distinguish  biofilm formation from early surface colonization.